# Adverse Renal Effects of Anticancer Immunotherapy: A Review

**DOI:** 10.3390/cancers14174086

**Published:** 2022-08-23

**Authors:** Maciej Borówka, Stanisław Łącki-Zynzeling, Michał Nicze, Sylwia Kozak, Jerzy Chudek

**Affiliations:** Department of Internal Medicine and Oncological Chemotherapy, Faculty of Medical Sciences in Katowice, Medical University of Silesia, Reymonta 8, 40-027 Katowice, Poland

**Keywords:** cancer, immunotherapy, adverse events, immune checkpoint inhibitors, chimeric antigen receptor therapy, bispecific antibodies, toxicity, renal, oncology

## Abstract

**Simple Summary:**

The immune system has a natural ability to work against cancer cells; however, in many cases this ability is insufficient, and cancers develop methods enabling them to escape from the supervision of immune cells. Novel therapeutic methods used in neoplastic diseases are based on encouraging immune cells to fight against cancer. In some cases, boosted by this approach, the immune system may damage not only tumor cells, but also other cells, tissues and organs in the human body. Kidney involvement, for example, is directly dangerous for patients’ health and may have an impact on human body homeostasis and the excretion of xenobiotics. However, renal function impairment in patients treated with immunotherapy is thought to be relatively rare but may be severe. Knowledge of early diagnosis and proper management are essential for physicians utilizing immunotherapy in daily clinical practice.

**Abstract:**

Modern oncological therapy utilizes various types of immunotherapy. Immune checkpoint inhibitors (ICIs), chimeric antigen receptor T cells (CAR-T) therapy, cancer vaccines, tumor-targeting monoclonal antibodies (TT-mAbs), bispecific antibodies and cytokine therapy improve patients’ outcomes. However, stimulation of the immune system, beneficial in terms of fighting against cancer, generates the risk of harm to other cells in a patient’s body. Kidney damage belongs to the relatively rare adverse events (AEs). Best described, but still, superficially, are renal AEs in patients treated with ICIs. International guidelines issued by the European Society for Medical Oncology (ESMO) and the American Society of Clinical Oncology (ASCO) cover the management of immune-related adverse events (irAEs) during ICI therapy. There are fewer data concerning real occurrence and possible presentations of renal adverse drug reactions of other immunotherapeutic methods. This implies the need for the collection of safety data during ongoing clinical trials and in the real-life world to characterize the hazard related to the use of new immunotherapies and management of irAEs.

## 1. Introduction

Epidemiologists predict approximately 3.4 million new cases of diagnosed cancers in the European Union (EU) and European Free Trade Association (EFTA) countries and 1.9 million in the USA, which may become the cause of about 1.7 million and 410 thousand deaths a year in 2040, respectively [1,2]. All of these make malignant neoplastic diseases a very important field of research, leading to the development of novel therapies, which have been continuously improving patients’ survival. Being very promising, this dynamic situation results in the necessity for physicians to learn how to deal with patients treated for cancer and how to manage long-term side effects caused by treatment [3].

Immunotherapy is believed to be one of the most popular and promising therapeutic approaches for cancer patients. This method is derived from the observation that cancer cells can escape from the control of the immune system and evade destruction by immunocompetent cells [4,5]. A few mechanisms, such as the production of immunosuppressive factors (e.g., TGF-β) by tumor cells [6,7] and the recruitment of cells that can mitigate immunological response [8,9,10], are considered to underlie this phenomenon. Reversing these effects and boosting the natural immune system is the key principle of immunotherapy [11]. There are various ways to achieve this, and they are evolving over time, starting from brave trials to induce erysipelas in patients with inoperable sarcomas to reduce tumor size by William B. Coley at the end of the 19th century [12,13]. Other invented strategies based on immunological response include the use of therapeutic cancer vaccines, oncolytic viruses and cytokines [14,15,16]. A milestone for present-day therapies was the research on cytotoxic T cell antigen 4 (CTLA-4) and programmed death receptor 1 (PD-1) proteins by James P. Allison and Tasuku Honjo, respectively, who were awarded the Nobel Prize in Physiology or Medicine in 2018 [17,18,19]. Since then, the era of immune checkpoint inhibitors (ICIs) has started. The advances in numerous different branches of medicine enabled the invention of chimeric antigen receptor T cells (CAR-T) therapy, which consists in collecting T cells from a patient’s peripheral blood, placing chimeric antigen receptors in collected cells via genetic engineering methods and then transferring them back to the appropriately prepared patient [20,21].

In common with other cancer treatments, such as chemotherapy, radiotherapy and surgery, methods based on mobilizing the immune system to destroy neoplastic cells are not free from adverse drug reactions. In the case of ICIs, they are specific to these therapeutic regimens and are called immune-related adverse events (irAEs). Damaging healthy cells by an agitated immune system lies at the root of them. Early recognition and adequate management of irAEs are crucial for patients’ safety and therapeutic success; therefore, being familiar with them is essential for physicians dealing with oncological patients [22]. Immune-related adverse events may affect every organ and system, especially the skin, the gastrointestinal tract, and the nervous, and endocrine systems [23,24]. Nephrotoxicity after ICIs is thought to be a relatively rare complication, but it may be underreported [25,26]. Since early identification of renal injury plays a meaningful role in a patient’s outcome, being aware of its possible manifestations and their management is a vital part of a physician’s knowledge.

## 2. Materials and Methods

In order to identify the literature concerning renal immune-related adverse events we conducted a search in the PubMed, Medline and Google Scholar databases. In the search process the following terms were used: “immunotherapy”, “immune-related adverse events”, “nephrotoxicity”, “renal”, “acute kidney injury”, “immune checkpoint inhibitors”, “CTLA-4”, “PD-1“, “PD-L1”, “trastuzumab“, “pertuzumab“, “panitumumab“, “cetuximab“, “rituximab“, “obinutuzumab“, “ofatumumab“, “brentuximab“, “alemtuzumab“, “IL-2“, “interferon“, “CAR-T cell”, “cancer vaccines”, and “bispecific monoclonal antibody”. Manuscripts were reviewed for titles, abstracts, and the entire text based on the following criteria: (1) original papers; (2) reviews; (3) renal immune-related adverse events as a key topic of the paper. The exclusion criteria were as follows: (1) methodological studies, editorials, commentaries, letters, and hypotheses; (2) no available abstract; and (3) manuscripts in a language other than English. The analysis was conducted in the following steps. The first step was related to the analysis of selected papers based on titles and abstracts, the second step was connected with the analysis of full-text papers, and the last step included the analysis of the collected data.

## 3. Immune Checkpoint Inhibitors (ICIs)

### 3.1. Mechanism of Action

The human immune system is not helpless in its confrontation with cancer cells. Not only does it fight infections that may lead to tumorigenesis, but it also recognizes and gets rid of suspicious cells in a process called immunosurveillance [27].

Immune cells identify neoplastic cells via neoantigens, defined as the proteins absent in healthy cells, which were produced in a process of transcription and translation of changed DNA sequence in cancerous cells [28]. The most important role in the recognition and further activation of immunological response involves antigen presenting cells (APCs), such as dendritic cells (DCs), which collect neoantigens, process and transfer them to secondary lymphoid organs where the antigen presentation takes place [29,30,31]. It happens due to the displaying of properly prepared tumor antigens via the major histocompatibility complex (MHC) present on the surface of the APC, which is then recognized by the T cell receptor (TCR) present on the surface of the T cells [32]. Complete activation of T cells is only possible if co-stimulation takes place. It is done by the interaction of the following proteins: CD28 on T cells and CD80/CD86 on APCs [33]. The proliferation of T cells then takes place, stimulated by autocrine or paracrine production of cytokines, especially interleukin 2 (IL-2) secreted by T cells [34]. Finally, activated T cells infiltrate the cancer tissue and recognize previously presented neoantigens, which enables the destruction of cancer cells. Neoantigens released by dead neoplastic cells amplify the immunologic response and therefore fulfill the cancer-immunity cycle [35,36].

The mechanism of T cell activation, however advantageous in terms of eliminating cancer cells, requires precise control to avoid excessive stimulation, which may result in damaging healthy tissues. The precise balance between optimal and excessive immune stimulation is maintained as a consequence of the interaction of special surface proteins called immune checkpoints, which takes place during the crosstalk between APC and T cells [37,38]. One of these proteins expressed on the surface of T cells, namely CTLA-4, competes with CD28 for binding with CD80/CD86. When the binding takes place, the signal for the proliferation of lymphocytes is suppressed [39,40]. The other important interaction, which weakens immunosurveillance, is the interplay of PD-1 with PD-L1 and programmed death-ligand 2 (PD-L2). PD-1 is present on the surface of the active T cells, whereas PD-L1 is expressed either on APCs or tumor cells [41,42]. All of these make a promising target for cancer therapies because suppressing inhibitory signals may improve the immune system’s capability to eradicate neoplastic cells [43]. This group of drugs, which actually are immunomodulatory monoclonal antibodies (mAbs), has been intensively studied and is still developing. The group of CTLA-4 inhibitors is represented by ipilimumab, while PD-1 is blocked for example by nivolumab and pembrolizumab [44]. Targets for particular ICIs are presented in Table 1.

### 3.2. Possible Manifestations and Pathophysiology of Renal irAEs

Blockade of immune checkpoints enhances patients’ immune cells’ capability to detect foreign cells, which simultaneously results in the possibility of classifying their own cells as foreign ones. This lies at the basis of irAEs that may occur during therapy with ICIs and that may affect almost every system of the human body [45]. The estimated total incidence of irAEs varies among different studies between 15% and 90% [46]. What is important is that irAEs may be recognized even several months after their administration [47]. These AEs may be mild to life-threatening or even result in death and are classified in five grades according to the Common Terminology Criteria for Adverse Events (CTCAE), where fifth grade means death [48]. Renal irAEs are less common than those involving the skin, lungs, bowels, liver, or endocrine glands [49]. Their frequency is estimated at up to 2% of cases [23], albeit some researchers anticipate that acute kidney injury (AKI) may occur even in 29% of cases [50,51]. Of note, AKI is less common during monotherapy than while combining two ICIs [52].

Clinically, renal toxicities may present as AKI, proteinuria, and dyselectrolytemia. There are also numerous possible types of renal injury after administration of ICIs, but acute tubulointerstitial nephritis (ATIN) is the most frequent one [53]. Other types include lupus-like immune complex glomerulonephritis [54], minimal change disease (MCD) [55,56], membranous nephritis (MN) [57], focal segmental glomerulosclerosis (FSGS) [58] and thrombotic microangiopathy (TMA) [59].

The exact mechanism leading to systemic or organ injury during (or after) therapy with ICIs is still unclear and necessitates further studies. Four possible mechanisms were proposed leading to renal irAEs [60]. The first one embraces the fact of expression of immune checkpoint molecules such as PD-L1 in kidneys, which may protect healthy tissue from T cell infiltration and cytotoxicity. Therefore, blockade of PD-L1 may result in tissue damage [60,61,62]. Another mechanism concerns activated T cells that can infiltrate either normal tissue or tumor and, in both cases, recognize antigens by TCRs. In the first case, TCR binds to antigens expressed on healthy cells that sequences are similar enough to neoantigens [60]. The next proposed mechanism involved in kidney injury is extensive production of pro-inflammatory cytokines such as IL-1Ra, CXCL10 and TNF-α [63], but it is still unclear whether increased levels of serum cytokines are a cause or an effect of tissue damage [60]. Last but not least, ICIs may contribute to the synthesis of different autoantibodies damaging normal organs [60]. In terms of the kidneys, there is a described case of anti-double-stranded DNA antibodies occurring in a patient’s blood after administering ICIs [64].

Some investigators concluded that tubulointerstitial nephritis caused by ICIs presents some differences from the classical ATIN caused by other drugs. Draibe et al. compared 13 patients with renal injury after taking ICIs with 34 patients with tubulonephritis related to other drugs and suggested that patients with ATIN related to ICIs had lower serum creatinine levels at the time of diagnosis (3.8  ±  1.0  vs. 6.0  ±  4.1 mg/dL, *p* < 0.01), and that the time from starting the treatment with the responsible drug to the diagnosis was longer in this group (197  ±  185 vs. 114  ±  352 days, *p* < 0.01) [65]. This suggests a milder course of kidney damage caused by ICIs.

### 3.3. Risk Factors

In a cohort study including 309 patients who were given ICIs and where 51 of them (16.5%) developed AKI, Meraz-Muñoz et al. performed the identification of risk factors for ICI-induced nephrotoxicity. The presence of hypertension (OR 4.3; 95%CI: 1.8–6.1), and cerebrovascular disease (OR 9.2; 95%CI: 2.1–40), administration of angiotensin-converting enzyme inhibitors/angiotensin-receptor blockers (OR 2.9; 95%CI: 1.5–5.7), diuretics (OR 4.3; 95%CI: 1.9–9.8) and corticosteroids (OR 1.9; 95%CI: 1.1–3.6), and other irAEs (OR 3.2; 95%CI: 1.6–6.0) predicted development of AKI in an univariate analysis. However, the multivariable analysis revealed an association only with hypertension (OR 2.96; 95%CI: 1.33–6.59) and other irAEs (OR 2.82; 95%CI: 1.45–5.48) [66].

Cortazar et al. in a multicenter study with 138 patients receiving ICI therapy found a lower estimated glomerular filtration rate (eGFR) (OR 1.99; 95%CI: 1.43–2.76), usage of proton pump inhibitors (PPIs) (OR 2.38; 95%CI: 1.57–3.62) and combination of anti-CTLA-4 with anti-PD1/anti-PD-L1 drug (OR 2.71; 95%CI: 1.62–4.53) to be risk factors of AKI [67].

Similarly, another cohort study, which included 429 patients treated with ICIs and 429 control patients, confirmed that PPIs administration (OR 2.40; 95%CI: 1.79–3.23) and the presence of other irAEs (OR 2.07; 95%CI: 1.53–2.78) are risk factors of AKI in patients treated with the mentioned type of immunotherapy [68].

Of note, PPIs are known to be able to induce interstitial nephritis manifested by AKI. It is estimated that omeprazole may induce acute interstitial nephritis in 2–20/100,000 treated patients [69,70]. Their impact on AKI development in patients treated with ICIs was an object of interest in numerous studies. Apart from the research mentioned above, such an association was also documented in other studies [71,72]. Risk factors for the ICI-induced nephrotoxicity are shown in Table 2.

### 3.4. Occurrence and Specific Nephrotoxicities

The first ICI to draw attention to possible renal adverse drug reactions was ipilimumab, an anti-CTLA-4 drug. In 2009 Fadel et al. noticed the possible harmful effects of ipilimumab on the kidneys. They reported a case of a 64-year-old man with metastatic melanoma who developed nephrotic syndrome after the treatment with this anti-CTLA-4 drug. The renal biopsy suggested lupus nephritis and anti-double-stranded DNA antibodies were detected. The treatment with ipilimumab was discontinued and prednisone was administered. After 3 months, anti-double-stranded DNA antibodies were undetectable and the nephrotic syndrome subsided [64]. In 2014, Izzedine et al. presented two case reports of patients with metastatic melanoma treated with ipilimumab with deteriorated kidney excretory function. In both cases a renal biopsy was performed and revealed interstitial inflammation. Both patients received prednisone administered orally and subsequently their kidney function improved [73].

In 2015 Thajudeen et al. described, as they claimed, the first case of biopsy-proven granulomatous interstitial nephritis after ipilimumab in a 74-year-old man with metastatic melanoma. The patient received treatment consisting of ipilimumab and dacarbazine. After the third cycle of therapy, the patient’s serum creatinine level doubled from 1.1–1.2 mg/dL to 2.2 mg/dL. Additionally, the patient complained of a rash. When the diagnosis was established based on biopsy, the treatment was interrupted and prednisone was applied. After 6 weeks kidney function improved. Finally, treatment with ipilimumab was resumed and the renal AE did not occur again [74].

Cortazar et al. in their work collected and summed up 13 cases of AKI after treatment with ICIs. Ten out of 13 patients were treated with ipilimumab alone or in combination. The period from starting the treatment to the development of AKI varied from 21 to 245 days with a median of 91 days. The median serum creatinine measured in these patients was 4.5 mg/dL. Seven patients had other irAEs recognized before the onset of AKI. All these patients had kidney biopsies performed. In 12 cases the histological diagnosis was ATIN and in one case it was TMA. Most of the patients (10) were treated with glucocorticoids and nine of them improved their renal function after the treatment. The remaining one, whose renal function did not recover after glucocorticoids, was the one with TMA. Patients who did not receive glucocorticoid therapy also did not improve their kidney function [52].

The ImmuNoTox study identified 14 ICI-induced AKI cases in 13 patients, retrospectively analyzing medical data from 352 patients treated with ICIs in one medical center in France. In most cases, the renal injury was classified as stage 1 (43%) and none of the patients needed hemodialysis therapy. Ten (77%) of these patients presented with irAEs affecting other systems. Six patients had renal biopsies which showed tubulointerstitial nephritis in all cases. The ICI therapy was withheld in all these patients and half of them received glucocorticoids. This study had some limitations related to its retrospective character [25].

It is worth remembering the fact that ICI-induced AKI was described not only after the treatment with ipilimumab but after other ICIs as well. There were also reported cases of nephrotoxicities associated with pembrolizumab [52,75,76] and nivolumab [77,78,79].

As far as pembrolizumab, an anti-PD-1 monoclonal antibody, is concerned, Izzedine et al. described a series of renal AEs in patients treated with this drug in one medical center. The authors observed a cohort consisting of 676 patients treated with pembrolizumab. In 12 participants (1.77%) renal side effects were observed, in 10 it was AKI, and in two proteinuria. In all mentioned cases of renal side effects, the kidney biopsy was performed, revealing acute tubular injury (ATI) in five patients, AIN in four patients, MCD alone in one patient, ATI and MCD in one patient, and finally nonspecific changes in one patient. In 10 patients pembrolizumab was withdrawn and seven of them received glucocorticoids. In one patient dialysis was started and this patient died in one month due to the progression of neoplastic disease. Others treated with glucocorticoids restored their renal function by about 50%. In one patient the treatment with pembrolizumab was restarted and resulted in an AIN relapse which was more severe. In patients who were not treated with glucocorticoids, the renal function remained stable. In two patients, in whom the treatment with pembrolizumab was maintained, their renal function improved [75].

In cases of biopsy-proven nephrotoxicity caused by nivolumab, the following findings occurred in the histological diagnosis: ATIN, IgA nephropathy, diffusive tubular injury, and complex-mediated glomerulonephritis. In the majority of the described cases, glucocorticoids were used in the treatment of these AEs resulting in renal function recovery [79].

Less frequent were renal AEs in patients treated with atezolizumab, durvalumab, avelumab and cemiplimab. Renal AEs were described for the first time for atezolizumab in a patient treated for renal cancer who developed AKI. The patient complained of an elevated body temperature and mild diarrhea. His blood tests revealed an elevation of serum creatinine to 5.6 mg/dL (in the previous tests the serum creatinine level was about 1.2 mg/dL). What is more, the urine tests showed proteinuria. This patient had a renal biopsy performed in which AIN was found. In the treatment methylprednisolone was used. The patient’s clinical improvement was observed after 8–10 weeks with partial normalization of serum creatinine level to 1.45 mg/dL [80]. In terms of durvalumab, there was a presented case of a patient who developed a nephrotic syndrome with MCD confirmed in a histological examination. The patient was treated with prednisolone and his symptoms withdrew [81]. In a phase II trial that included 88 patients treated with avelumab for chemotherapy-refractory metastatic Merkel cell carcinoma, four episodes of AKI occurred [82]. There is also reported a case of AKI with biopsy-proven AIN in a patient treated for squamous cell carcinoma of the skull with cemiplimab. The patient reported weakness and fatigue. His serum creatinine level was elevated in comparison to previous results (2.87 mg/dL and 1.3 mg/dL, respectively). The patient was treated with glucocorticoids and his renal function improved. After 3 months his serum creatinine level was 1.47 mg/dL [83]. Renal AEs caused by treatment with ICIs are summarized in Table 3.

The real incidence of AKI induced by ICIs is a subject of vivid debate. Cortazar et al. investigated the data from phase II and III clinical trials, with 3695 patients who were receiving ICIs. AKI occurred in 2.2% and severe AKI, defined as an increase of serum creatinine to a level higher than 4 mg/dL or tripling of initial creatinine level, emerged in 0.6% of patients. Furthermore, the incidence of AKI differed between the patients treated with various ICIs, ranging from 1.4% for pembrolizumab, 1.9% for nivolumab, and 2.0% for ipilimumab to 4.9% for combined therapy with ipilimumab and nivolumab [52]. However, some authors suggest the real incidence of ICI-induced AKI may be much higher with a range of 9.9–29% [50].

### 3.5. Management and Outcomes

Due to more and more frequent use of ICIs in cancer treatment and still rising awareness of possible renal adverse effects of the mentioned therapy, both the ESMO and ASCO included recommendations on renal toxicities management in their guidelines concerning immunotherapy.

ESMO guidelines divide patients with nephritis related to ICIs into four grades depending on serum creatinine elevation in relation to its baseline or upper limit of normal (ULN) (G1: 1.5 × baseline or >1.5 × ULN; G2: 1.5–3 × baseline or >1.5–3 ULN; G3: 3 × baseline or >3–6 ULN; G4: >6 × ULN) and recommend different strategies in each group. In general, according to these guidelines, every patient should have their serum sodium, potassium, creatinine, and urea level checked before each ICI application. In the case of abnormalities, other causes of renal function impairment such as dehydration, infection or obstruction in the urinary tract should be taken into account and then systematically excluded. What is more, potentially nephrotoxic drugs should be withdrawn. As for serious disturbance in renal parameters, ICI therapy should be suspended and administration of glucocorticoids should be considered (0.5–2 mg/kg/day methylprednisolone or equivalent). In dubious situations, renal biopsy may be taken into consideration as well as nephrology consultation [84]. Of note, urea level measurement is not recommended by nephrological guidelines for the diagnosis of AKI [85].

In the ASCO guidelines patients with worsened renal function caused by ICI therapy are also divided into four groups, but based on slightly different criteria which include direct creatinine elevation instead of elevation over ULN (G1: 1.5–2 × above baseline or >0.3 mg/dL; G2: 2–3 × above baseline; G3: >3 × above baseline or >4.0 mg/dL; G4: 6 × above baseline). Similarly to the ESMO guidelines this paper also highlights the importance of looking for other causes of kidney function deterioration. Diagnosis of ICI-induced AKI is empirical, and the biopsy is not indicated in most cases except for the ones not susceptible to standard treatment. In addition, the ASCO in the first line of treatment suggests glucocorticoids in doses of 0.5–2 mg/kg/day prednisone equivalents. Interestingly, they mention using other immunosuppressive drugs such as infliximab, azathioprine, cyclophosphamide, cyclosporine A, and mycophenolate in cases refractory to glucocorticoids [86]. The possible utility of infliximab [87] and mycophenolate [88,89] in the treatment of irAEs after ICI regimens was suggested in a few scientific papers.

In the analysis performed by Cortazar et al. which included 138 patients with ICI-induced AKI, 40% had complete renal function recovery, while 45% had partial recovery and 15% did not improve their renal function after treatment. These patients were treated in 86% of cases with glucocorticoids and in 97% their ICI therapy was suspended. The therapy was resumed in 31 patients with previously diagnosed ICI-induced AKI and AKI relapsed in seven patients out of 31 (23%) [52]. An observational cohort study published by Baker et al. showed that AKI was associated with higher mortality in the group of patients treated with ICIs (HR 2.28; 95%CI: 1.90–2.72) but patients with AKI related to ICI therapy had significantly lower mortality (HR 0.43; 95%CI: 0.21–0.89) than patients with the other causes of AKI [90].

## 4. Tumor-Targeting Monoclonal Antibodies (TT-mAbs)

In general, mAbs are a very large and composite group of drugs, which may be divided into immunomodulatory and TT-mAbs. The first subgroup has already been discussed quite extensively in the previous section by the example of ICIs. Here, we briefly characterize the second subgroup.

### 4.1. Mechanism of Action

There are several mechanisms where neoplastic cells may be affected by TT-mAbs. Generally, most frequently they are able to inhibit signaling pathways essential for both the survival and progression of cancer cells, which takes place as a result of modifying the receptor proteins function [91,92] or binding to specific tumor-associated antigens (TAA) [91,93]. Another common feature for a significant part of TT-mAbs is their ability to promote antibody-dependent cell-mediated cytotoxicity (ADCC) [94,95] or complement-dependent cytotoxicity [96], which is strictly connected with the opsonization of malignant cells. The most popular and widely used anticancer drugs among TT-mAbs include: (1) anti-human epidermal growth factor receptor-2 (HER-2) mAbs such as trastuzumab, trasuzumab emtasine [T-DM1] and pertuzumab mainly in breast cancer [97,98] and gastric/gastroesophageal junction cancer [99]; (2) anti-epidermal growth factor receptor (EGFR) mAbs represented by panitumumab and cetuximab which are utilized in the treatment of colorectal [100] and head and neck cancers [101,102]; (3) anti-CD20 mAbs exemplified by rituximab, obinutuzumab and ofatumumab administered mostly in hematological malignancies such as chronic lymphocytic leukemia (CLL) [103] and non-Hodgkin lymphomas (NHLs) [104]; (4) anti-CD30 mAb brentuximab vedotin and (5) anti-CD52 mAb alemtuzumab, which are also used especially in different hematological malignancies [105,106]. A detailed discussion of the exact action mechanism of each of the mentioned TT-mAbs is beyond the scope of this work.

### 4.2. Renal Adverse Effects

Trastuzumab, an anti-HER-2 mAb, is known for its possible cardiotoxicity [107]. In a study comparing chemotherapy alone versus chemotherapy with trastuzumab, no significant difference in the occurrence of AKI in both groups was detected [108]. However, there is described a case report of a patient treated with ado-trastuzumab emtansine (T-DM1), a drug that consists of trastuzumab and maytansinoid DM1 which has cytotoxic properties, in whom nephrotic syndrome developed after the beginning of the therapy. In this patient, a renal biopsy revealed FSGS and ATI [109]. Worth mentioning is the fact that decreased kidney filtration existing before trastuzumab therapy may increase the risk of cardiotoxicity of this therapy [110]. In the case of the other anti-HER-2 mAb, pertuzumab, no important renal AEs were reported [111,112].

As far as anti-EGFR mAbs (panitumumab, cetuximab) are concerned, the most important renal AE of these therapies is dyselectrolytemia, in particular hypomagnesemia and hypokalemia [113,114]. A meta-analysis performed by Petrelli et al. showed that treatment with these mAbs may induce hypomagnesemia. The estimated incidence of lowered levels of serum magnesium (Mg^2+^) was 17% and is thought to be higher in the group treated with panitumumab than in patients treated with cetuximab [115]. Hypomagnesemia is believed to be caused by decreased activation of the renal EGFR, which results in lowered activation of the TRPM6 (transient receptor potential cation channel), leading to reduced reabsorption of Mg^2+^ [116]. In terms of hypokalemia in patients treated with cetuximab, hypokalemia of any grade was observed in 8% of patients [117], while in patients undergoing therapy with panitumumab hypokalemia of all grades was detected in 34% of patients [118]. The exact mechanism of hypokalemia is still not fully explained. Management of these electrolyte imbalances is based on watchful monitoring of patients at risk and proper supplementation of potassium and magnesium when needed [113]. Boku et al., in a post-marketing surveillance study assessing the safety of panitumumab in 3085 patients, found 12 (0.4%) renal and urinary disorders without further specifying its exact character [119]. There was also a case of nephrotic syndrome, AKI and leukocytoclastic vasculitis in a patient treated with panitumumab [120]. Furthermore, there were reported cases of AKIs and nephrotic syndromes in patients treated with cetuximab. Histopathological findings in these cases included crescentic diffuse proliferative glomerulonephritis [121], diffuse proliferative glomerulonephritis [122] and TMA [123].

The next group of mAbs contains an anti-cluster of differentiation 20 (anti-CD20) antibodies. The most common AEs caused by this group of mAbs are infusion-related reactions, infections and cytopenias caused by reversible myelosuppression [124,125,126]. In terms of kidneys, AKI may be caused by tumor lysis syndrome (TLS) which may occur after treatment with rituximab, obinutuzumab and ofatumumab; therefore, physicians supervising therapy with anti-CD20 agents should be aware of this possible side effect [127,128]. In addition, the infections mentioned above may affect the urinary tract [129].

The most common AEs in the course of treatment with anti-CD30 antibody conjugate with auristatin E include fatigue, nausea, diarrhea, neutropenia and peripheral sensory neuropathy [130]. In a study comparing the efficacy and safety of treatment with brentuximab vedotin versus treatment with pembrolizumab, in one (0.7%) patient out of 152 receiving brentuximab vedotin ATIN occurred, while in this group AKI or nephritis was not spotted [131].

Side effects of alemtuzumab, anti-CD52 mAb in most cases manifest as an influenza-like syndrome, transient cytopenias and increased susceptibility to infections [132]. In a study designed to assess the safety of therapy for CLL with alemtuzumab in which 149 patients received alemtuzumab, no particular nephrotoxicity was spotted [133]. However, in patients treated with alemtuzumab because of non-oncological indications, for multiple sclerosis, there were several cases of anti-glomerular basement membrane (anti-GBM) disease and membranous glomerulonephropathy [134]. Interestingly, there was noted a case of prostate and kidney aspergillosis in a patient treated for CLL connected with the immunosuppressive properties of alemtuzumab [135]. Renal AEs caused by treatment with TT-mAbs are summarized in Table 4.

## 5. Chimeric Antigen Receptor T Cell (CAR-T Cell) Therapy

### 5.1. Mechanism of Action

CAR-T cell therapy is a particular example of adoptive T-cell therapy (ACT) which generally relies on using a patient’s own T cells to destroy cancer cells. These T cells need to be previously collected and then properly modified to enable them to recognize abnormal cells [136]. As far as CAR-T cells are concerned, preparing them starts with collecting the patient’s peripheral blood and further isolation of T cells using leukapheresis. Then T cells proliferated, and CARs are placed in their cell membrane using molecular biology techniques [137]. CARs are transmembrane proteins that consist of an extracellular part that binds to the selected antigen, a spacer/hinge part, a transmembrane part and an intracellular one that is involved in signal processing and T cell activation [138,139,140]. Following such a preparation, CAR-T cells are re-infused into patients’ circulation after the administration of the lymphodepleting chemotherapy [141]. Binding of CAR to the targeted antigen activates effector functions of T cells independently from MHC [142]. Activation of T cells induces the production of cytokines or cytotoxic activity with expected anti-cancer effects [143].

This type of therapy is mainly used in patients with refractory or resistant hematological malignancies such as B cell acute lymphoblastic leukemia (B-ALL) and diffuse large B-cell lymphoma (DLBCL). Nowadays, there are more and more attempts to use CAR-T cells to fight solid tumors [144,145,146,147].

### 5.2. Renal Adverse Effects and Their Pathomechanisms

One of the main limitations of CAR-T cell therapy is its toxicity, often severe and life-threatening. Predominantly AEs of this therapy include cytokine release syndrome (CRS) and neurotoxicity, also called CAR T-cell related encephalopathy syndrome (CRES) [148]. CRS is associated with a massive production of cytokines in response to the binding of CAR to the targeted antigen and following activation of the immune response. Main cytokines involved in CRS include IL-6, IL-10 and interferon (IFN-γ) [148,149,150,151]. Symptoms and severity of CRS vary among patients, starting from influenza-like symptoms to the dysfunction of almost all organs and systems [152,153,154]. There are some grading systems used to assess the severity of CRS [155,156].

Renal AEs include AKI related to CRS [157], related to prerenal and renal mechanisms [158]. Prerenal AKI after CAR-T treatment is associated with impaired renal perfusion caused predominantly by CRS complications such as fever or vomiting, which may lead to dehydration resulting in a reduction in the intravascular volume [159]. In addition, severe CRS may lead to vasodilation, capillary leak syndrome and reduction of cardiac output. All of these affect kidney perfusion and result in a decrease in the glomerular filtration rate [160,161]. The renal mechanisms of AKI are also the consequences of prolonged hypovolemia leading to tubular ischaemic injury [162] and direct tubular toxicity of cytokines [163,164]. As a result of treatment and damage to neoplastic cells, tumor lysis syndrome (TLS) may develop. This syndrome is caused by the release of the contents from destroyed cells, inter alia intracellular ions, nucleic acids, proteins, and their metabolites. Substances such as uric acid and phosphate may contribute to the damaging of renal tubules when they precipitate, which in consequence leads to renal function impairment [165,166]. Other possible mechanisms of nephrotoxicity of CAR-T therapy are associated with the consequences of the development of hemophagocytic lymphohistiocytosis (HLH) in which AIN or TMA may be identified [167,168,169].

### 5.3. Occurrence and Outcomes

In a study performed by Gupta et al., researchers evaluated the incidence of AKI in 78 patients treated with axicabtagene ciloleucel (YESCARTA^®^) or tisagenlecleucel (KYMRIAH^®^) for refractory DLBCL. Among 15 patients (19%) with AKI, eight of them had lowered kidney perfusion, six developed ATN and one had an urinary tract obstruction related to the progression of the lymphoma. In this study, grade 3 of AKI was confirmed in six patients and three of them required kidney replacement therapy. However, the average length of hospitalization and 60-day mortality was similar in patients with and without AKI [170].

Gutgarts et al. analyzed data from 46 adult patients treated for Non-Hodgkin lymphoma (NHL) with axicabtagene ciloleucel (YESCARTA^®^) or tisagenlecleucel (KYMRIAH^®^). They assessed kidney function up to 100 days after initiation of the treatment. They reported AKI of any grade in 14 (30% of patients) and grade 2 or 3 in 4 (8.7%) patients. In this study, none of the patients required kidney replacement therapy and most of them recovered kidney function within 30 days [171].

Another study included 38 patients treated with tisagenlecleucel (KYMRIAH^®^) for DLBCL. In this study AKI was diagnosed in two (5%) patients; both of them had grade 3 AKI. One of them died 4 days after treatment and the second one 28 days after treatment [172].

Finally, a systematic review performed by Kanduri et al., based on 22 cohort studies including 3376 patients treated with CAR-T cells, revealed an 18.6% (95%CI: 14.3–23.8) incidence of AKI, while 4.4% (95%CI: 2.1–8.9) of patients required renal replacement therapy [173].

### 5.4. Management

Patients who underwent CAR-T cells therapy and developed AKI should have been treated for the cause of renal function impairment. In the case of a prerenal mechanism, patients with hypovolemia should receive proper fluid resuscitation and vasopressors when needed. In such cases, norepinephrine is the first-choice drug for these patients [174]. Those with a clinically significant deterioration in cardiac output should be considered to be candidates for inotropic agents such as milrinone, dopamine, epinephrine, norepinephrine, or vasopressin [175].

Patients who developed AKI in the course of CRS or HLH should receive supportive care and proper treatment for these disease entities. ASCO guidelines cover these issues in detail [176]. The use of tocilizumab—an anti-human interleukin-6 receptor (anti-IL-6R) monoclonal antibody (mAb)— or in some cases of CRS, may be beneficial [177]. Treatment of HLH is based on immunosuppression with glucocorticoids, IL-6 antagonists or etoposide [176,178]. When it comes to TLS, patients at risk should be identified before therapy and proper precautions should be taken. Hydration and administration of hypouricemic drugs such as allopurinol or rasburicase should be considered [179].

## 6. Therapeutic Cancer Vaccines

### 6.1. Mechanism of Action

Tumor vaccines are another approach in cancer treatment based on attempts to mobilize the immune system to fight against cancer. In contrast to prophylactic vaccines such as human papillomavirus (HPV) and hepatitis B vaccines, which are known to prevent cervical and hepatocellular cancer, respectively [180,181], therapeutic cancer vaccines are used for inducing tumor regression, eliminating minimal residual disease, and initiating the establishment of the immunological memory. Generally speaking, tumor vaccines make use of the natural mechanism in which antigens deriving from the cancer cells are uptaken by DCs, which migrate to the peripheral lymphoid organs, where antigen presentation on MHC I and MHC II takes place. It is how naive CD4+ and CD8+ T cells are engaged in anti-cancer activity. Regarding the fact that cancers can suppress natural immunity, in many cases this mechanism is ineffective. The vaccines with properly prepared exogenous antigens and often in the company of adjuvants that facilitate DCs activation try to restore repressed immunity [182,183]. For now, there are three therapeutic vaccines approved by the Food and Drug Administration (FDA) for use in the therapy: intravesical Bacillus Calmette–Guérin (BCG) live in early-stage bladder cancer, sipuleucel-T (PROVENGE^®^) in metastatic castration-resistant prostate cancer and talimogene laherparepvec (IMLYGIC^®^) in metastatic melanoma [184]. The exact mechanisms of action of these agents differ from each other and are complex; therefore, discussing them is beyond the scope of this study.

### 6.2. Renal Adverse Effects

Most of the described renal AEs associated with therapeutic cancer vaccines were caused by intravesical BCG live. This form of therapy has been approved by the FDA since 1990. Regarding the intravesical way of administration of BCG, the most common side effects are local with cystitis and bacterial infection, whereas systemic adverse drug reactions may include elevated body temperature and fatigue [185]. Although AEs of intravesical BCG administration are typically mild, there were also some described cases with a severe course [186]. Among immunological complications, polyarthritis is the most frequent one [187]. Renal AEs were also reported, which are believed to be relatively rare. The majority of described cases referred to AKI caused by interstitial nephritis. In these cases, the therapy with glucocorticoids was administered and in three out of eight patients the renal function recovered [188,189,190]. The other single cases include nephritis in the course of Henoch–Schönlein purpura [191] and membranous glomerulonephritis with clinical presentation of nephrotic syndrome [192]. It is worth mentioning that during follow-up after the intravesical BCG therapy some asymptomatic kidney lesions may be found in the imaging diagnosis, which may turn out to be kidney granulomas. Regarding the scarcity of evidence, the proper management of this entity has not been determined yet. Some authors suggested using anti-tuberculous drugs [193], while others did not recommend such therapy [194]. Further studies in this matter might be useful.

In terms of sipuleucel-T (PROVENGE^®^), there are some data about AEs which come from the FDA Adverse Event Reporting System (FAERS). Sipuleucel-T (PROVENGE^®^) was approved by the FDA in 2010 and there were 3216 AEs reported which covered 9600 patients treated with this therapeutic cancer vaccine from 2010 to 2017. The majority of the reported AEs involved elevated body temperature, shivers, malaise, and fatigue. Interestingly, in this paper researchers described 48 cases of hematuria and 24 cases of hydronephrosis [195]. On the other hand, in the clinical trial including 512 patients and 341 treated with sipuleucel-T (PROVENGE^®^), no renal toxicity was identified [196].

The third vaccine—talimogene laherparepvec (IMLYGIC^®^) was approved by the FDA in 2015 for the treatment of melanoma [197]. In the OPTiM trial, 162 patients were treated with this vaccine and similarly to the previously mentioned therapies most common AEs were chills, elevated body temperature, influenza-like symptoms, and fatigue. In the group of irAEs, two cases of glomerulonephritis—the first one with renal papillary necrosis and the second one followed by acute renal failure were reported [198]. Considering that cases of renal injury in patients treated with talimogene laherparepvec (IMLYGIC^®^) were rarely described, there is not sufficient information about its exact pathomechanism and proper management.

## 7. Bispecific Monoclonal Antibody (BsAb)

### 7.1. Mechanism of Action

Bispecific monoclonal antibodies (BsAbs) are antibodies with two different binding sites. Each of these binding sites can bind to different antigens or different epitopes of the same antigen. The possible application of BsAbs is not limited only to immunotherapy of hematological and oncological malignancies, but these antibodies may be used also in the treatment of neovascular age-related macular degeneration, Alzheimer’s disease, rheumatoid arthritis, and other entities [199,200]. Several BsAbs action mechanisms may be applicable in oncology. The first of them is connected with the blocking of two molecular pathways at the same time [201]. The next one is based on the concept of blockade of two different immune checkpoints [202]. And the most popular strategy uses one binding site of the antibody to bind to the tumor-associated antigen (TAA) and the other binding site to bind to the molecule on the immune cell, in most cases it is CD3 on T cells. Such BsAbs are called Bi-specific T-cell engagers (BiTEs). When TAA on the neoplastic cell and CD3 on the T cell are bound together, a structure called cytolytic synapse is created and then the T cell releases enzymes such as granzyme B and perforin, which contribute to the destruction of the tumor cell [203,204].

BsAbs are a very promising therapeutic approach in numerous medical fields; therefore, they are intensively studied. A number of different combinations of binding sites have been tested so far and probably more and more BsAbs will receive FDA or European Medicines Agency (EMA) approval over time. There are currently two BsAbs approved by the FDA for use in the treatment of malignancies. The first drug is blinatumomab (Blincyto^®^) which can bind to CD3 on T cells and to CD19 on B cells and is used in the treatment of Philadelphia chromosome-negative (Ph-) relapsed or refractory B-cell precursor acute lymphoblastic leukemia (ALL) [205,206]. The second one is amivantamab-vmjw (Rybrevant^®^), which targets epidermal growth factor receptor (EGFR) and mesenchymal epithelial transition factor (MET) and can be used in the treatment of non-small cell lung cancer (NSCLC) [207]. Additionally, mosunetuzumab (Lunsumio^®^), being an anti-CD3 and anti-CD20 BsAb, is conditionally approved by the EMA for use in relapsed or refractory follicular lymphoma (FL) [208,209].

### 7.2. Renal Adverse Effects

The safety of blinatumomab (Blincyto^®^) was assessed in clinical studies, suggesting that almost all patients experience AEs associated with such treatment and the majority of them (68–87%) have AEs of grade 3 or more severe. The most frequent AEs were pyrexia, headache, and edema. One of the possible negative consequences of therapy with this drug was the development of CRS—with a frequency of approximately 4.9–12%. In these studies, neither renal function deterioration nor an increased serum creatinine level was reported [210,211,212,213].

In terms of mosunetuzumab (Lunsumio^®^) in clinical studies, treatment-related AEs emerged in 74.1% of patients. The most common ones were neutropenia, CRS, hypophosphatemia, fatigue, and diarrhea. In the early trials, no renal AEs such as AKI or serum creatinine elevation were reported [214]. Regarding the scarcity of the data, further studies on this matter may be needed.

## 8. Cytokine Therapy

### 8.1. Mechanism of Action

As for the treatment with cytokines, these are cell signaling molecules which have an autocrine and paracrine activity, allowing them to influence immunological response regulation [215]. In view of the foregoing, some cytokines found a use not only for the therapy of noncancerous diseases (e.g., hepatitis B [216] and hepatitis C [217] or Behçet’s syndrome [218]), but nowadays they also constitute a part of anticancer immunotherapy because of their ability to inhibit the growth of neoplasm due to the mediation of immune-nonimmune cell interactions in the tumor microenvironment [219]. Indeed, only Interferon-alpha (IFN-α) and Interleukin-2 (IL-2) have been so far approved by the FDA for cancer treatment as monotherapy. IFN-α was approved for the treatment of follicular lymphoma [220], hairy cell leukemia [221], melanoma [222] and Kaposi’s sarcoma associated with AIDS [223], while IL-2 was approved for the treatment of advanced renal cell carcinoma (RCC) [224] and metastatic melanoma [225]. Explaining the exact action mechanism of each of the mentioned cytokines in detail is beyond the scope of this paper.

### 8.2. Renal Adverse Effects

Considering AEs of treatment with IFN-α, the most common are influenza-like symptoms, including elevated body temperature, shivers, myalgia, headache and nausea. These symptoms appear in the majority of patients treated with this cytokine. Other AEs of administering IFN-α comprise, among others, hematological toxicities, loss of appetite and therefore loss of body mass, but also elevated liver enzymes levels, depressed mood and chronic fatigue [226,227]. Regarding nephrotoxicities, it is estimated that the most common renal AE is proteinuria and the transient elevation of serum creatinine, which may be present in as many as 10–25% of patients undergoing therapy with IFN-α [228,229]. More severe renal side effects, including nephrotic syndrome and AKI, were also described [230]. Described pathological findings in patients with AKI include AIN, ATN, FSGS, MCD, and TMA [231,232,233].

Among the most common AEs of the second cytokine used in the cancer treatment, namely IL-2, are increased body temperature, malaise, nausea, liver enzyme levels elevation and hematological abnormalities. Worth mentioning also is capillary leak syndrome (CLS), which may also occur in the course of treatment with IL-2 [234]. In CLS, serum proteins are lost from intravascular space due to increased capillary permeability, which leads to reduction of volemia, peripheral oedemas, effusions in serous cavities and in some cases to shock [160]. Elevated serum creatinine levels, oliguria and decreased sodium excretion are thought to be relatively common in patients treated with IL-2 and may be present in more than 60% of patients receiving IL-2 treatment. These findings are in most cases associated with decreased renal blood flow caused by, for instance, hypovolemia. The management should be concentrated on restoring normal renal perfusion and avoiding other nephrotoxic factors [235,236]. There is also some evidence that renal mechanisms may be involved in kidney function impairment as well in patients treated with IL-2 [237]. Feinfeld et al. described a case of a patient treated with IL-2 who developed AKI with pathological features of AIN [238].

## 9. Future Directions

Regarding the fact that renal AEs are relatively rare complications of immunotherapy and numerous promising treatments are in the phase of research, creating a registry of renal irAEs is urgently needed.

In terms of ICIs and their renal AEs, future research may be trying to determine if ATIN caused by ICIs differs from ATIN caused by other drugs and whether this difference implicates changes in the optimal treatment [65]. The real incidence of this type of AE is undetermined and awareness of the possibility of renal AEs occurrence may facilitate finding the real prevalence of renal adverse drug reactions in patients treated with ICIs [50,52]. There are also many unanswered questions concerning the proper management of AIN, especially the most beneficial model of the treatment with glucocorticoids, their dosage, method of administration, and treatment duration [52]. The possible role of drugs such as infliximab [87] and mycophenolate [88,89] in the treatment of AIN caused by ICIs requires further elucidation.

As far as renal AEs associated with CAR-T cell therapy are concerned, their real occurrence and exact mechanisms leading to renal function impairment should be clarified [113,125,126,127,128]. Efficient methods of their management should be found and the possible application of tocilizumab in this indication requires evaluation [131,132].

Better insight into the renal irAEs of cancer vaccines is also needed. The utility of the administration of anti-tuberculous drugs in the case of kidney granuloma developed in the course of intravesical BCG therapy is a crucial issue [148,149]. When it comes to sipuleucel-T (PROVENGE^®^) and talimogene laherparepvec (IMLYGIC^®^), available data about renal AEs of these therapies are really poor and this matter should be consecutively explored [151,153].

Renal adverse drug reactions of BsAbs are hardly known [165,166,167,168,169]. Considering the large number of ongoing trials assessing the safety and efficacy of different new BsAbs [160], their impact on kidneys should be described soon.

## 10. Conclusions

The development of innovative anti-cancer therapies utilizing interactions with the immune system is very promising in terms of improving the outcomes of patients with various malignancies. These anti-cancer immunotherapies are boosting the patient’s natural immunological mechanisms, which contribute to the destruction of neoplastic cells by the components of the immune system. However, in common with other therapeutic approaches immunotherapy causes AEs. Most of them are associated with extensive excitation of the immune system which causes damage not only to the neoplastic cells but also to the host’s healthy tissues in every organ and system in the human body, including the kidneys. Therefore, being familiar with them is an important element of knowledge for physicians dealing with oncological patients. Taking into account the impact of kidneys on maintaining the body homeostasis and its role in the metabolism of xenobiotics, this group of AEs demands rapid recognition and proper management. Renal AEs are generally thought to be rare, but their prevalence is probably underestimated. The most widely studied renal AEs in this group of drugs are those caused by ICIs. AKI, ATIN, proteinuria, or dyselectrolytemia occur in up to 2% of patients treated with ICIs. As for the ESMO and ASCO guidelines, they contain few recommendations concerning the management of these and other types of irAEs. Proper hydration and avoiding nephrotoxic drugs are indicated in all cases. Administration of glucocorticoids should be considered in the severe clinical course of renal irAEs. Most serious ones required cessation of ICIs therapy. Deterioration of renal function during CAR-T cell therapy may be caused by various mechanisms and the management should be focused on the removal of factors leading to kidney injury. The data about renal AEs caused by therapeutic cancer vaccines and BsAbs are fragmentary and incomplete. Collection of safety data in clinical trials and real-life data will show the hazard related to the use of new immunotherapies.

## Figures and Tables

**Table 1 cancers-14-04086-t001:** Summary of Immune Checkpoint Inhibitors (ICIs).

Target	Drug
TLA-4	Ipilimumab (Yervoy^®^) Tremelimumab *
PD-1	Nivolumab (Opdivo^®^) Pembrolizumab (Keytruda^®^) Cemiplimab (Libtayo^®^) Dostarlimab (Jemperli^®^)
PD-L1	Atezolizumab (Tecentriq^®^)
	Avelumab (Bavencio^®^) Durvalumab (Imfinzi^®^)

CTLA-4: cytotoxic T-lymphocyte-associated protein 4, PD1: programmed cell death protein 1, PD-L1: programmed death-ligand 1. * not yet approved.

**Table 2 cancers-14-04086-t002:** Risk factors for the ICI-induced nephrotoxicity.

Risk Factor	AEs	OR	95%CI	*p*-Value
The presence of other irAEs	AKI [66]	3.2	1.6–6.0	<0.001
	AKI [68]	2.07	1.53–2.78	No data
Hypertension	AKI [66]	4.3	1.8–6.1	<0.001
Cerebrovascular disease	AKI [66]	9.2	2.1–40	<0.001
ACEI/ARB	AKI [66]	2.9	1.5–5.7	<0.01
Diuretics	AKI [66]	4.3	1.9–9.8	<0.001
Corticosteroids	AKI [66]	1.9	1.1–3.6	<0.05
eGFR < 30 mL/min/1.73 m^2^	AKI [67]	1.99	1.43–2.76	<0.001
PPIs	AKI [67]	2.38	1.57–3.62	<0.001
	AKI [68]	2.40	1.79–3.23	No data
Anti-CTLA-4 with anti-PD1/anti-PD-L1 combination	AKI [67]	2.71	1.62–4.53	<0.001

AEs: adverse events, OR: odds ratio, CI: confidence interval, irAEs: immune-related adverse events, AKI: acute kidney injury, ACEI: angiotensin-converting-enzyme inhibitors, ARB: angiotensin receptor blockers, eGFR: estimated glomerular filtration rate, PPIs: proton pump inhibitors, Anti-CTLA-4: anti-cytotoxic T-lymphocyte-associated protein 4, Anti-PD1: anti-programmed cell death protein 1, Anti-PD-L1: anti-programmed death-ligand 1.

**Table 3 cancers-14-04086-t003:** ICI-induced nephrotoxicity.

Drug	Target	AEs
Ipilimumab (Yervoy^®^)	CTLA-4	Nephrotic syndrome [64]
		Interstitial inflammation [73]
		Granulomatous interstitial nephritis [74]
		Acute interstitial nephritis [52]
		Thrombotic microangiopathy [52]
		Tubulointerstitial nephritis [25]
Pembrolizumab (Keytruda^®^)	PD-1	Acute tubular injury [75]
		Minimal change disease [75]
Nivolumab (Opdivo^®^)	PD-1	Acute tubulointerstitial nephritis [79]
		IgA nephropathy [79]
		Diffusive tubular injury [79]
		Complex-mediated glomerulonephritis [79]
Atezolizumab (Tecentriq^®^)	PD-L1	Acute interstitial nephritis [80]
Durvalumab (Imfinzi^®^)	PD-L1	Nephrotic syndrome [81]
		Minimal change disease [81]
Cemiplimab	PD-L1	Acute interstitial nephritis [83]

AEs: adverse events, CTLA-4: cytotoxic T-lymphocyte-associated protein 4, PD1: programmed cell death protein 1, PD-L1: programmed death-ligand 1.

**Table 4 cancers-14-04086-t004:** TT-mAb-induced nephrotoxicity.

Drug	Target	AEs
Ado-trastuzumab Emtansine (Kadcyla^®^)	HER-2	Nephrotic syndrome [109]
		Focal segmental glomerulosclerosis [109]
		Acute tubular injury [109]
Panitumumab (Vectibix^®^)	EGFR	Hypomagnesemia [113,114]
		Hypokalemia [118]
		Renal and urinary disorders [119]
		Nephrotic syndrome [120]
		Acute kidney injury [120]
		Leukocytoclastic vasculitis [120]
Cetuximab (Erbitux^®^)	EGFR	Hypomagnesemia [113,114]
		Hypokalemia [117]
		Nephrotic syndrome [120]
		Acute kidney injury [120]
		Crescentic diffuse proliferativeglomerulonephritis [121]
		Diffuse proliferative glomerulonephritis [122]
		Thrombotic microangiopathy [123]
Rituximab (Mabthera^®^)	CD20	Acute kidney injury [127,128] Urinary tract infections [129]
Obinutuzumab (Gazyvaro^®^)		
Ofatumumab (Kesimpta^®^)		
Brentuximab vedotin (Adcetris^®^)	CD30	Acute tubulointerstitial nephritis [131]
Alemtuzumab (Lemtrada^®^)	CD52	Anti-glomerular basement membrane disease [134]
		Membranous glomerulopathy [134]
		Kidney aspergillosis [135]

AEs: adverse events, HER-2: human epidermal growth factor receptor 2, EGFR: epidermal growth factor receptor, CD20: B-lymphocyte antigen CD20, CD30: tumor necrosis factor receptor superfamily member 8, CD52: cluster of differentiation 52.

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
