# Peer review of "Adverse Renal Effects of Anticancer Immunotherapy: A Review"

_cancers, 2022, doi:10.3390/cancers14174086_

Round 1
Reviewer 1 Report
This is an interesting manuscript on renal adverse effects of anticancer immunotherapy. Authors mainly discussed four different types of anticancer immunotherapies and associated renal adverse effects, putative mechanisms, occurrence and outcomes, management and outcomes. Overall, this is a very valuable study, however, there are some concerns which prevent the acceptance of it under current form.
-
Authors described that “There are scarce data concerning renal adverse drug reactions of other immunotherapeutic methods.” Is it because there are limited patients or clinic data associated other immunotherapeutic methods or less renal adverse effect were found associated with other immunotherapeutic methods?
-
line 115, should it be “PD-L1” instead of “P-L1”? in the sentence “… with P-L1 and programmed death-ligand 2 (PD-L2)…”?
-
The risks factors are only discussed for ICIs not for other three types of anticancer immunotherapies. What are specific reasons?
-
It seems tables are not referred in content.
-
In Table 2, there are two entries of “PPIs/AKI”, it would be necessary to label that they are from different references. There are three entries of “PPIs/Interstitial nephritis leading to AKI”, what are the differences among these three entries and what is the purpose of listing them if no data are present.
Reviewer 2 Report
In this review the authors describe adverse renal effects upon anticancer immunotherapy. They collcted available information on different immunotherapies and summarized them in a well structured and very comprehensive way. Thereby they highlight the need for specific precautions in terms of renal survaillance during anticancer immunotherapies.
The review is well written and summarizes the latest reserearch on the field. In the end they show future directions and classify the information in the overall context.
I just have some minor comments concerning the content of this work.
1. As mentioned before the structured description of the single therapies is very convenient. However the information on pathomechanisms and managment of the bispecific antibodes as well as cancer vaccines seem to be missing or very short. Is this due to lacking information or due to redundances. Maybe the authors could adress this shortly in the text
2. The authors have chosen several, by no doubt, important immunological intervention strategies for cancer treatment. It is just not clear to me, what was the decision strategy behind this ? For example why did the authors focus only on bispecific antibodies and did not include regular antibody therapies such as trastuzumab. Furthermore, it would be interesting to see cytokine therapy or Parp inhibition in the context of renal damage. It could be nice to, at least, address these shortly in the text or comment on the selection of the described therapies.
3. To my knowledge there are also immunotherapies used for treatment of renal cancer. Is there a special need for precautions of additional renal damage in this case or do the incidence remain similar to other cancer entities.
